# How Regulation 536/2014 Is Changing Academic Research with Therapeutic Radiopharmaceuticals: A Local Experience

**DOI:** 10.3390/ph18111709

**Published:** 2025-11-11

**Authors:** Valentina Di Iorio, Stefano Boschi, Erika Brugugnoli, Maddalena Sansovini, Federica Matteucci, Carla Masini, Manuela Monti

**Affiliations:** 1IRCCS IRST Dino Amadori, Via Piero Maroncelli 40, 47014 Meldola, Italy; erika.brugugnoli@irst.emr.it (E.B.); maddalena.sansovini@irst.emr.it (M.S.); federica.matteucci@irst.emr.it (F.M.); carla.masini@irst.emr.it (C.M.); manuela.monti@irst.emr.it (M.M.); 2Department of Pharmacy and Biotechnology, University of Bologna, 40126 Bologna, Italy; stefano.boschi@unibo.it

**Keywords:** Regulation (EU) No. 536/2014, investigational therapeutic radiopharmaceuticals, GMP

## Abstract

**Background/Objectives**: This report examines the future of academic studies involving investigational therapeutic radiopharmaceuticals within the framework of Regulation (EU) No. 536/2014. It discusses the impact of Good Manufacturing Practice (GMP) requirements (EudraLex-Volume 4-Good Manufacturing Practice guidelines) on the development of radiopharmaceuticals, based on local experience and analysis. **Methods**: The report was drafted by analysing multiple factors, including the European regulatory context regarding EMA guidance for investigational medicinal products (IMPs) and GMP requirements for radiopharmaceuticals, as well as position papers from various scientific associations. An analysis of all the relevant changes was conducted by a multidisciplinary team comprising radiopharmacists, nuclear medicine physicians, research experts and technology transfer specialists. They conducted a literature review to examine the clinical implications of the regulatory change and assess the impact of Regulation 536/2014 on academic clinical trials. **Results**: IRST has around 20 years’ experience in radiopharmaceutical clinical research. From 2008 to 2025, it conducted 16 clinical trials with radiopharmaceuticals under the Directive, and it is currently promoting five studies under the Regulation. During this time, more than 1000 patients were enrolled. The transition was based on staff training in quality documentation, the constitution of a contract research organisation (CRO) to ensure data quality and transfer, careful budget planning, the evaluation of innovative business models and the role of a Contract Development and Manufacturing Organization (CDMO). These integrated approaches enabled IRST to transform regulatory constraints into an opportunity to enhance its organisational model, improve data reliability, and strengthen its position as a centre of excellence for radiopharmaceutical research and production. **Conclusions**: The implementation of EU Regulation 536/2014 has significantly impacted academic research centres, especially those specialising in radiopharmaceuticals. Adhering to Good Manufacturing Practice (GMP) for therapeutic radiopharmaceuticals requires a considerable investment in infrastructure and personnel. However, the regulation also presents opportunities for research centres to enhance their capabilities. Meeting GMP standards can help institutions improve the quality and reliability of their clinical trials, potentially making them more competitive in the international research arena.

## 1. Introduction

The European regulatory landscape for clinical trials has undergone a significant transformation with the introduction of Regulation (EU) No. 536/2014 [1], which replaced the previous Directive 2001/20/EC. This regulatory shift represents a transition from a fragmented, nationally implemented framework to a centralised, directly applicable one. The new regulation primarily aims to harmonise and streamline the approval processes and conduct of clinical trials across EU member states.

One of the most impactful changes introduced by Regulation (EU) No. 536/2014 is the requirement for compliance with Good Manufacturing Practice (GMP) guidelines (EudraLex–Volume 4) when producing investigational therapeutic radiopharmaceuticals.

Specifically, Article 61 states that the manufacture of investigational medicinal products within the Union must comply with GMP. The only exception for radiopharmaceuticals pertains to diagnostic investigational medicinal products when the manufacturing process is carried out in hospitals, health centres, or clinics by pharmacists or other authorised personnel in the relevant Member State. This exception applies only if the investigational medicinal products are intended exclusively for use within the same hospitals, health centres, or clinics participating in the same clinical trial in that Member State. As a result, GMP compliance for diagnostic radiopharmaceuticals remains voluntary and at the discretion of each Member State, leading to considerable variation across Member States [2,3].

Historically, radiopharmaceutical research has primarily been conducted by non-profit academic institutions, often targeting rare diseases and orphan indications. However, recent advancements in molecular imaging, the growing demand for precision oncology solutions, and emerging data from international literature have significantly increased commercial interest in this field. Consequently, the radiopharmaceutical sector is evolving from a niche academic field into a commercially viable industry with a focus on profitability [4].

Mandatory authorisation and GMP compliance for investigational therapeutic radiopharmaceuticals have had a significant impact on the field of radiopharmaceuticals, as many of these agents were historically produced in academic institutions. Indeed, many theranostic agents were traditionally produced ‘in-house’ by academic centres without authorised GMP facilities. The GMP requirement poses sustainability challenges for academic and non-profit institutions, which often lack the necessary infrastructure and financial resources to meet these regulatory demands independently.

Setting up GMP-compliant facilities requires significant initial and ongoing expenditure, often amounting to millions of dollars, with costs per square foot increasing in line with the complexities of pharmaceutical manufacturing processes. The increasing complexity and cost of GMP-compliant radiopharmaceutical production is prompting the sector to adopt collaborative models with pharmaceutical companies and contract research organisations (CROs). Consequently, theranostic development is shifting towards a more commercially driven landscape, moving away from small-scale, orphan-targeted research, such as studies on neuroendocrine tumours, towards large-scale programmes addressing high-burden cancers, such as prostate cancer. Furthermore, integrating diagnostic imaging with targeted therapeutic strategies constitutes a significant advancement in precision medicine, generating increased interest from the pharmaceutical industry.

Since 2008, IRST (Istituto Romagnolo per lo Studio dei Tumori “Dino Amadori”) has promoted independent clinical research in the field of radioligand therapy, establishing itself as a reference centre for rare diseases such as neuroendocrine tumours, for which there are few available therapeutic options.

Very few commercial therapeutic radiopharmaceuticals were available for cancer treatment until 2017 i. This meant that patients with no other therapeutic options could only be treated in clinical trials promoted by the few centres in Italy capable of producing radiopharmaceuticals in-house.

This production was permitted by national legislation, which allowed hospital production to be exempt from the GMP requirements for pharmaceutical factories (Article 16, D. Lgs. 200/2007).

In this context, over the past ten years, IRST has offered treatment to more than 100 patients per year in non-profit clinical trials, attracting more than 80% of patients from outside the region.

However, this regulatory framework has now been substantially amended by Regulation (EU) No. 536/2014, which imposes an authorisation requirement for the manufacture of investigational medicinal products for both profit and non-profit clinical trials.

This regulation came into force in January 2025 due to the transition period, as well as the launch of the Clinical Trials Information System (CTIS) website, which took several years to develop.

Since 2014, our institute has considered setting up a new facility to produce investigational therapeutic radiopharmaceuticals and ensure patient access to these treatments. Drawing on our local experience, this article examines how Regulation (EU) No. 536/2014 is reshaping academic research in therapeutic radiopharmaceuticals by considering the interplay of regulatory, technical, economic and clinical factors, thereby ushering in a new era for translational academic research.

## 2. Results

### 2.1. Overview of the Regulatory Context

Regulation (EU) No. 536/2014 was published in the *Official Journal of the European Union* in 2014 but did not come into force until January 2025. This delay allowed nuclear medicine centres time to prepare for the Regulation’s major challenges. The impact of the GMP requirements for investigational therapeutic radiopharmaceuticals was clearly underestimated by academic centres, and only those that acted promptly to implement the necessary changes were able to comply with the Regulation’s timelines. The greatest impact was on academic centres that had been producing investigational therapeutic radiopharmaceuticals in-house in accordance with national legislation (e.g., Article 16 of D. Lgs. 200/2007 and the ‘Norme di Buona Preparazione dei Radiofarmaci per la Medicina Nucleare’ (NBP-MN) in Italy), and which are now required to establish GMP-compliant facilities. This has not had a particularly disruptive impact on the international scientific community, as these are niche therapies and the number of centres producing them was certainly limited. Furthermore, several European countries, including Germany and Spain, have a greater number of authorised academic facilities to produce radiopharmaceuticals. Consequently, these countries have already developed more robust infrastructures, whereas Italy was much further behind in complying with GMP standards.

Italy is one of the few Member States to grant a GMP exemption for investigational therapeutic radiopharmaceuticals in a non-profit setting. Therefore, the repeal of D. Lgs. 200/2007 had a particularly strong impact in Italy, as hospital-produced therapeutic radiopharmaceuticals were no longer exempt from GMP. In other countries, no such distinction existed and academic GMP standards were already widely applied prior to Regulation 536 [5].

Associations such as the Italian Association of Nuclear Medicine (AIMN) and the European Association of Nuclear Medicine (EANM) have also played an important role in raising awareness of this issue among regulatory bodies [6,7,8]. Several articles have been published in the international scientific community on the impact of Regulation 536 on academic research [2]. Since 2014, experts have been evaluating the implications of the new regulation for academic radiopharmaceutical production [2,5], and it was widely believed that the regulation would force most academic centres to close as compliance with GMP would require excessive investment.

As shown in Table 1, IRST began evaluating the potential impact of the new legislation in 2015 but only obtained GMP authorisation in 2024. This long-time interval was due to several factors, including internal feasibility analyses, interactions with the regulatory agency (AIFA), the technical timelines required to apply for competitive calls to support the investment, the authorisation processes for constructing the facility, radioactivity-related authorisations, submitting clinical trials to both the Ethics Committee and AIFA, and personnel qualifications. The activation of the site was further delayed by about 1 year due to the impact of the pandemic and a flood in the Emilia-Romagna region.

To date, our centre is the only academic institution in Italy with GMP authorisation for investigational therapeutic radiopharmaceuticals. AIFA has provided continuous support throughout the development of this pioneering project, playing a significant role in achieving GMP certification. Early engagement with regulatory authorities, discussions with other national and international nuclear medicine centres, interactions with opinion leaders, and collaborations with scientific societies made the successful completion of the GMP authorisation process possible.

#### 2.1.1. Key Findings in the Transition from Directive 2001/20/EC to Regulation (EU) No. 536/2014

Implementing Regulation (EU) No. 536/2014 presented a significant challenge to clinical research centres, especially those specialising in investigational therapeutic radiopharmaceuticals. IRST is a clinical reference centre for treatment with investigational radiopharmaceuticals. From 2008 to 2025, IRST promoted ten clinical trials with therapeutic radiopharmaceuticals and six clinical trials with diagnostic radiopharmaceuticals under the Directive. Currently, IRST is promoting five active studies under the Regulation (three for therapy and two for diagnostics). However, the activation rate of IRST-sponsored therapeutic radiopharmaceutical trials decreased by 50% during the transition from the Directive to the Regulation. Since 2019, patient enrolment has gradually declined due to the market availability of Lutathera^®^ and Pluvicto^®^, an increasing number of company-sponsored trials and a reduction in IRST-sponsored studies. Enrolment in therapeutic studies ranged from 60 to 80 new patients per year under the Directive, compared with 30–40 per year under the Regulation. Notably, around 70% of enrolled patients came from outside the Emilia-Romagna region. Overall, since 2008, IRST has treated over 1000 patients with therapeutic radiopharmaceuticals. Regarding diagnostic studies, the annual number of new patients enrolled has been around 300 since 2015, with a decline since 2022 related to market availability and/or the inclusion of radiopharmaceuticals in the European Pharmacopoeia monograph.

Below are discussed the main issues that IRST has encountered in transitioning to the new regulatory framework:Authorisation procedures: the transition from the national ‘Osservatorio sulle sperimentazioni cliniche’ to the Clinical Trials Information System (CTIS) has introduced several layers of complexity. The increased administrative burden, stricter timelines, new rules governing ethics committees in Italy and the need to comply with radioprotection requirements—further complicated by recent legislative updates—have made the implementation of the Regulation considerably more challenging.

The average authorisation time under the Directive and the Regulation is similar (4–6 months).

In general, we have not observed significant differences in timelines for IRST-sponsored studies in the transition from the Directive to the Regulation when the study is single-centre; however, this difference becomes much more evident when the study is multicentre. In the specific case of therapeutic radiopharmaceuticals, the three ongoing studies were migrated from the Osservatorio alle Sperimentazioni Cliniche (OsSC) to the Clinical Trials Information System (CTIS), so the timelines are not directly comparable. The major difference lies in the significant administrative burden and the larger amount of documentation required by the Regulation compared to the Directive. To handle more documentation, infrastructure staff must be well-trained and able to meet strict regulatory deadlines.

Contract research organisation (CRO) constitution: under the Regulation, management of a study by a non-profit organisation is considered equivalent to management by a for-profit sponsor. This has significantly increased the workload of academic research infrastructures in terms of both documentation requirements and staff expertise. Therefore, IRST has self-certified as a CRO with AIFA to manage studies from conception to final approval, including submission to ACs and ECs, monitoring, CRFs, quality controls, statistical analyses, and so on. This ensures data quality and the ability to transfer the data and results of non-profit clinical trials for registration purposes, which could reduce the time it takes for new treatments to reach patients. IRST CRO staff have expertise in scientific, statistical, regulatory and administrative aspects, and two people have been trained in CTIS, with one person dedicated to pharmacovigilance. Approximately 20 people work in the CRO, including monitors who perform monitoring visits as required by GCP.Administrative and financial implications for investigator-initiated trials: the increase in workload has also led to higher management costs for clinical studies, which must necessarily be covered by dedicated research funds. IRST has carefully developed a budget planning system, strengthening the application for national and European grants to ensure the sustainability of academic research. The cost of managing a study under the Regulation compared with the Directive has increased by 40%, excluding drug-related costs, which will be addressed in more detail below.Interactions with companies that supply precursors and intellectual property considerations: the promotion of IRST studies aimed to ensure patient access to treatments for which there were no alternatives available on the market. Preparing and acquiring technical documentation for submission to ECs and ACs often required close interaction with companies that owned the precursors, particularly the cold component of the radiopharmaceutical. In most cases, this component was proprietary. This raised issues regarding confidential company data, intellectual property management and pharmacovigilance. In such cases, ad hoc contracts were drafted to address these issues. Most of IRST’s radiopharmaceutical studies involve drafting a collaboration/supply agreement with pharmaceutical or biotech companies. These agreements also included provisions governing pharmacovigilance and supply.Innovative business model evaluations: the new Regulation enabled new forms of collaboration between public and private stakeholders, such as co-partnering and co-sponsorship, to ensure the sustainability of research [9]. Co-sponsorship encourages collaboration among researchers and institutions, facilitating knowledge sharing and joint research efforts. IRST can serve as a CDMO for early-phase (1–2) radiopharmaceutical production or as a centralised radiopharmacy for regional or national multicentre studies. IRST has already established a public–private collaboration agreement with a biotech company for the GMP production of a drug in a Phase 1 study involving a radiopharmaceutical. This agreement was the first of its kind and will serve as a model for future collaborations. Developing it required integrating multiple areas of expertise, including legal, pharmaceutical-technical, regulatory and economic competencies.

Thanks to these integrated approaches, IRST was able to turn regulatory constraints into an opportunity to improve its organisational model, enhance data reliability and strengthen its position as a leading centre for radiopharmaceutical research and production.

#### 2.1.2. Competition Across EU Member States

Although Regulation (EU) 536 standardises research across Europe, there are still national challenges. This is due to the high complexity of radiopharmaceuticals. In addition to the regulatory authority responsible for approving clinical trials, several non-harmonised bodies deal with radioprotection aspects, which often delay the final approval process. In Italy, for instance, the Ministry of Health/ISS requires an additional 30 days of silent consent. Such discrepancies create heterogeneity among Member States, making the management of international studies involving radiopharmaceuticals more challenging in terms of both manufacturing and authorisation. Furthermore, the implementation of Regulation (EU) 536 in Italy has had a negative impact on centres producing therapeutic radiopharmaceuticals under the Directive. Italy is comparatively more disadvantaged than other Member States in this respect, given that academic institutions abroad have long had the capacity to manufacture under GMP conditions. During the first public consultation held in the second half of 2021, the European Association of Nuclear Medicine (EANM) emphasised that the regulatory framework to produce in-house radiopharmaceuticals is not harmonised throughout Europe. This has resulted in uneven access to innovative radiopharmaceuticals based on the particularities of national legislation [10].

### 2.2. Overview of Technical Aspects

We shared our Institute’s decision to set up a new GMP facility with the National Regulatory Authority at two separate events: an open AIFA meeting and a dedicated online meeting. During the dialogue, AIFA emphasised that the IRST facility could provide a platform for IRST and other academic centres to centralise the production of experimental therapeutic radiopharmaceuticals in accordance with EU 536/2014.

The project to build a new GMP-compliant radiopharmacy was incorporated into a broader regional initiative focusing on the ‘Centralised Compounding Centre (CCC) for Centralised Oncology Pharmacy and Radiopharmacy’. The aim of this project was to establish an innovative compounding centre where oncological and radiopharmaceutical therapies could be prepared centrally in response to the needs of the Romagna area. The project’s mission was to reduce costs by improving the efficiency, quality, and safety of production lines.

Construction of the new building was completed at the end of 2023, and validation activities were performed in the first half of 2024 to submit a request to AIFA for authorisation to produce experimental radiopharmaceuticals according to EudraLex-Volume 4-Good Manufacturing Practice (GMP) guidelines.

The execution of these activities, including qualification, validation and improvement of the Quality System, required regulatory support from Good Manufacturing Practice (GMP) experts. The application to activate the new facility was submitted on 20 June 2024, and the IRST facility was inspected by the National Regulatory Authority (AIFA) from 5 to 8 August 2024, in compliance with Article 111(5) of Directive 2001/83/EC. Based on the findings of the inspection, AIFA deemed the IRST facility to comply with the Good Manufacturing Practice requirements set out in Directive 2003/94/EC, and granted manufacturing authorisation no. aM-132/2024 on 5 September 2024, in accordance with Article 40 of Directive 2001/83/EC. Figure 1 summarises the main steps taken by IRST for the transition to the new Regulation and the activation of the new Facility.

The GMP authorisation relates to the production of ‘Investigational radiopharmaceuticals based on lutetium-177 in pharmaceutical form: Small Volume Liquids Prepared Under Aseptic Conditions’.

From January to October 2025, the IRST GMP facility, also known as the IRST Radiopharmaceutical Therapy Factory (RTF), has provided 83 batches of experimental therapeutic radiopharmaceuticals labelled with Lu-177, totalling 219 patient doses.

#### 2.2.1. General Costs

Excluding building-related costs (construction and setup), the true sustainability challenge lies in maintaining authorisation and ensuring efficient production.

Table 2 summarises the main cost categories involved in the routine GMP production of experimental radiopharmaceuticals and the corresponding proportion of total operational budget.

#### 2.2.2. Radionuclide Cost

The approval of new treatment options, such as [^177^Lu]Lu-oxodotreotide-Lutathera^®^ for neuroendocrine tumours and [^177^Lu]Lu-vipivotide tetraxetan-Pluvicto^®^ for prostate cancer, has increased demand for lutetium and other radionuclides, which could exceed the current supply. Consequently, the price of lutetium-177 is rising due to high demand, as well as the complex, high-flux neutron facilities and specialised reactors required for its production. Recent procurement tenders in Italy indicate that the cost of lutetium-177 has doubled from 2024 to 2025.

The shutdown of European research reactors without immediate replacement capacity, coupled with the general ageing of infrastructure, remains an unsolved issue requiring concerted European action [11]. This poses serious challenges for European radionuclide producers.

European pharmaceutical legislation also requires marketing authorisation (MA) for radionuclides. This is a critical issue because the cost of radionuclides affects the budget for pharmaceutical healthcare [12]. Radionuclides are not medicinal products because they require labelling with specific ligands for therapeutic use. A revision of European pharmaceutical legislation is currently underway and could potentially address this issue [13].

#### 2.2.3. Production Capacity

Currently, the new Facility-RTF can meet IRST’s internal production requirements, as well as external requests from other centres.

According to the current authorisation, one manufacturing slot is available five days a week, with a capacity of four doses per slot, equating to 18–20 doses per week or around 900–1000 doses per year.

Following AIFA approval, radiopharmaceutical manufacture was reduced from six to four patient doses per batch due to the switch from manual to automated production, as needed by GMP regulations.

However, it is important to note that the production of investigational radiopharmaceuticals is closely associated with the volume of active clinical studies and trends in patient enrolment observed across clinical centres. RF production depends on active clinical trials, as these are investigational drugs. IRST’s challenge is to either find new radiopharmaceuticals to test or use known radiopharmaceuticals in new combinations or dosing schedules.

Since these are experimental drugs, the regulatory authority provides authorisation for each type of radionuclide rather than for each medicinal product. Given this, there can be different cases, which are detailed in Table 3.

#### 2.2.4. Sustainability

The CCC project received its initial funding through two tenders: one from the Emilia-Romagna Region (RER) in 2017, and another from the Ministry of Economic Development (MISE). Further financial support came from IRST funds and charitable donations.

Approximately 35% of the overall research funding was directed to the activation of the new Facility based on Good Manufacturing Practice (GMP) that included: building, implementation of new equipment, validation and personnel training. The remaining funds were used for the implementation of the new Centralized Oncology Pharmacy, which is located in the same building.

The IRST facility can produce investigational radiopharmaceuticals under its current authorisation. This is the only way to ensure patients have access to therapies not of interest to pharmaceutical companies. As investigational medicinal products (IMPs) are not granted marketing authorisation, they cannot be placed on the market and must instead be provided free of charge by the study sponsor. Therefore, the financial sustainability of IMP supply within clinical research can only be ensured through the allocation of national and international research funding specifically designated to cover the costs of new clinical trials. Alternatively, this can be achieved through structured partnerships and co-sponsorship agreements with private sector entities.

One model that could support the feasibility of academic trials with novel radiopharmaceuticals would be to implement multicentre studies based on a cost–recovery approach, whereby participating institutions contribute to the direct expenses of centralised production. Within this framework, the study would maintain its non-commercial nature, as financial contributions would be strictly limited to covering production costs. This approach could facilitate academic access to investigational radiopharmaceuticals while ensuring compliance with regulatory standards.

Nevertheless, this model must be set out in a comprehensive contractual agreement that is subject to review and formal approval by the relevant ethics committee and competent authority.

#### 2.2.5. Competition with the Company

Another constraint in academic research is the increasing interest of pharmaceutical companies in the radiopharmaceutical market. One example is [^177^Lu]Lu-DOTATATE. This radiopharmaceutical, which was developed through academic research and used for years in non-profit clinical trials, has now moved to pharmaceutical company as [^177^Lu]Lu-oxodotreotide (Lutathera^®^). Although Lutathera^®^ received centralised marketing authorisation from the European Medicines Agency (EMA), significant price disparities have been reported across EU Member States. These variations reflect differences in national pricing strategies, reimbursement frameworks and local manufacturing capabilities.

In the Netherlands, where [^177^Lu]Lu-DOTATATE was originally developed, the Ministry of Health has actively supported continued in-house (magistral) preparation of the radiopharmaceutical in hospital settings. This approach results in markedly lower costs than those associated with commercially supplied doses: approximately €4000 per hospital-produced patient dose versus around €23,000 per dose from pharmaceutical manufacturers, excluding an additional €2500 in transportation costs [11].

Recent trends indicate that pharmaceutical companies are showing a growing interest in the field of radiopharmaceuticals, driving and consolidating new lines of research. This development is closely related to the expansion of translational research, as well as to the progress of personalised medicine and targeted therapeutic approaches.

In this context, pharmaceutical companies are frequently acquiring biotechnology firms engaged in the discovery and development of novel ligands. However, such acquisitions can limit the availability of these compounds for independent, non-profit clinical trials conducted within academic settings. Consequently, innovations in radiopharmaceutical research are often integrated into the industrial pipeline at an early stage, and academic institutions only gain access once intellectual property rights have been secured by industry. IRST has found itself competing with companies, which significantly reduces the opportunities to promote new studies.

This evolving scenario has important implications for academic research, which is facing increasingly strong competition from industrial stakeholders with greater financial resources and more advanced technical expertise. However, in this context, the combination of clinical expertise and basic research remains a distinctive strength of academia and is essential for its continued contribution to the field’s development.

It should be emphasised that all the major innovations in nuclear medicine today, such as PSMA, SSR and FAPI, emerged because academics recognised the need to address unmet medical requirements [14]. This issue concerns not only the development of new ligands, but also the use of new radionuclides—particularly alpha emitters—for developing new radiopharmaceuticals, which are the focus of profit-oriented studies. IRST, therefore, faces the challenge of identifying new ligands or developing novel scientific questions that do not overlap with those already proposed by companies.

### 2.3. Overview of Clinical Scenario

Supply chain and isotope constraints remain a critical bottleneck, impacting trial design, scalability and clinical implementation. The transition from non-profit, small-scale production to profit-driven, large-scale production has also restricted academic centres’ ability to produce radiopharmaceuticals, as companies have secured intellectual property rights for each new product.

Historically, radiopharmaceuticals developed for rare diseases (e.g., neuroendocrine tumours) within academic settings were produced and administered in centres with extensive experience in handling these highly complex agents. Once pharmaceutical companies recognised the potential of such radiopharmaceuticals, industry investment in isotope production accelerated rapidly [15].

Herrmann et al. [16] reported that, in 2018 alone, more than 1000 publications were indexed in PubMed under the search terms ‘theranostic’ or ‘theragnostic’, reflecting the growing interest in radioisotope-based therapeutics (radiotheranostics). The rapid growth of industry-sponsored trials has significantly increased the number of patients being treated, leading to challenges in large-scale production due to the intrinsic characteristics of these agents. This development has significantly limited opportunities for academic institutions to conduct independent research.

Furthermore, Regulation (EU) No. 536/2014 imposed additional limitations on centres producing investigational radiopharmaceuticals under GMP exemptions; from 31 January 2025, they will no longer be permitted to manufacture experimental therapeutic radiopharmaceuticals according to national regulations. These two factors have substantially hindered academic research, which had previously played a crucial role in ensuring patient access to such therapies, particularly for rare diseases that fall outside the scope of commercial interest.

The scientific community and nuclear medicine societies (EANM and AIMN) issued recommendations regarding the implications for clinical practice [17], the identification of challenges in non-commercial research and the specific features of academic trials targeting rare cancers. They also discussed the potential of non-commercial studies in generating robust clinical evidence [5]. The need for centres of excellence to manage these treatments and perform multidisciplinary case reviews was strongly highlighted.

At IRST, multidisciplinary teams have been set up to cover all tumour types, with a particular focus on diseases that could potentially benefit from radiometabolic treatments, such as neuroendocrine and prostate tumours. These teams include oncologists, nuclear medicine physicians, palliative care specialists, radiologists, pharmacists, nurses and study coordinators. Regular meetings are held to coordinate patient care, discuss treatment strategies and review ongoing clinical trials. NET multidisciplinary meetings are held fortnightly and discuss around 8–10 cases, while prostate cancer meetings are held weekly and discuss 10–15 new cases.

Furthermore, the EMA published a concept paper addressing the clinical development of radiopharmaceuticals in oncology [18]. Societies specialising in nuclear medicine and oncology can play a significant role in supporting and strengthening academic research in the field of radiopharmaceuticals.

The IAEA has recently developed a theragnostic curriculum for nuclear medicine doctors and other physicians, paving the way for specialised training programmes in therapeutic radiopharmaceuticals [19]. As Scott et al. [19] report, this is an exciting time for the use of theranostics in oncology; however, the rapid growth of this area of nuclear medicine has also created challenges. In particular, the infrastructure for manufacturing and distributing radiopharmaceuticals is still being developed, and regulatory bodies are optimising guidelines for this new class of drugs.

There is a current trend towards industry-sponsored trials that focus not only on niche indications, but also on high-impact diseases such as prostate cancer, while also looking to expand indications for rare conditions wherever possible. There has been rapid expansion, particularly in radioligand therapy (RLT, theranostics) for prostate and neuroendocrine tumours, with several ongoing and planned trials, as well as several high-profile late-stage programmes. Major pharmaceutical companies have invested in and acquired capabilities, moving from early proof-of-concept studies to larger registration trials, with several phase III programmes recently underway. Theranostics and personalised medicine approaches are increasingly emphasised, driving trial designs that integrate imaging endpoints, biomarkers and adaptive strategies. In this evolving landscape of profit-driven research, the IRST has outlined new strategies to ensure patients without alternative treatment options have access to therapies, by promoting non-profit clinical protocols for indications outside of those that are approved.

A key feature of IRST-sponsored studies has been the anticipation of RLT in earlier disease settings and its combination with external beam radiotherapy. Other features include modifications to approved treatment schedules, such as dose reductions and shorter intervals between cycles. In line with current trends, future protocols will also examine the combination of RLT with systemic oncological therapies [20]. Another area of research being pursued by IRST involves locoregional applications, particularly in the treatment of breast cancer. Activating such trials has required IRST to address multiple challenges, including obtaining trial authorisation through CTIS, considering sustainability, and negotiating with companies that hold intellectual property rights for peptides. IRST is one of the few centres in Italy authorised to conduct phase I trials with radiopharmaceuticals. As of 10 July 2025, only eight centres nationwide have self-certified this capability.

As radioligand therapies have become more widely used in the treatment of cancer patients, a further challenge has emerged: incorporating RLT into treatment algorithms for these patients. This is evident in prostate cancer, for example, with the approval and commercialisation of [^177^Lu]Lu-PSMA-617 ([^177^Lu]Lu-vipivotide tetraxetan-Pluvicto^®^) but is less apparent in neuroendocrine tumours treated with [^177^Lu]Lu-DOTATATE ([^177^Lu]Lu-oxodotreotide- Lutathera^®^), where there are limited approved alternative therapies for patients. Consequently, oncologists and nuclear medicine physicians have encountered numerous challenges in developing shared pathways for managing oncology patients. In this context, the role of multidisciplinary groups has become increasingly critical. Close interaction between oncologists, radiotherapists, and nuclear medicine physicians has been demonstrated by the incorporation of RLT into standard patient care algorithms [21,22]. At IRST, multidisciplinary meetings have been significantly implemented for prostate cancer and neuroendocrine tumours, particularly given the large number of patients who come from outside the institute. A multidisciplinary team is essential, particularly as the optimal therapeutic pathway for these patients has not yet been clearly defined [23].

Following the implementation of Regulation 536, IRST was required to ensure GMP-compliant production to continue providing access to RLT for patients whose conditions do not fall within the authorised indications. As IRST is the only academic centre capable of providing GMP-grade production, many Italian centres now refer patients to IRST, as they can no longer guarantee access themselves. Consequently, around 60–70% of patients treated at IRST come from other regions. Access to RLT is further restricted by the limited availability of approved therapies, as regional authorities are forced to impose limits on the number of patients who can be treated for economic reasons.

This complex scenario has raised a series of ethical concerns for public healthcare centres, particularly for IRST, which has received numerous requests that could be met from a technical standpoint, but not from a regulatory or authorisation perspective.

Another ethical concern arises in relation to the growing problem of shortages, as companies are finding it increasingly difficult to sustain ever-expanding production.

Academic centres such as IRST could mitigate shortages by providing radiopharmaceutical production in emergency situations. To enable such an approach, interaction with regulatory authorities is essential.

The limited access to currently marketed drugs is related both to the lack of authorised dispensing centres in many regions and to the narrow target population defined in the product label, particularly in the case of neuroendocrine tumours.

These limitations could be overcome through the production of experimental radiopharmaceuticals for patients who do not fall within the approved indications. The promotion of multicentre studies across Italy, supported by a centralised radiopharmacy for production, could enhance access to these therapies for a larger number of patients.

Another relevant aspect for an academic centre is economic sustainability. IRST has had to address the increasing costs associated with GMP-compliant production, which are significantly higher than those incurred under national regulations.

As all treatments are experimental, patient access is only possible within the framework of clinical trials. In this case, the sponsor (IRST) must bear all the costs related to the trial. Costs outside of clinical practice cannot be passed on to the national healthcare system, so IRST must find funding to cover them. Until now, IRST has relied on national grant funding, which has only partially covered the costs. However, there is a need to strengthen grant applications to support the increasing costs of academic research [16].

One potential source of funding is to conduct research in collaboration with pharmaceutical companies, as Regulation 536 allows for new forms of public–private collaboration and data sharing for regulatory purposes. Regulation (EU) No. 536 provides new opportunities for academic studies, enabling these centres to generate evidence that profit-driven companies can use for regulatory purposes. Using data for registration purposes enhances the social and ethical use of the research. An Italian Ministerial Decree (30 November 2021) permits this use, provided it complies with the Regulation’s provisions. The context is complex and unpredictable, with a constantly evolving regulatory framework that needs to be tested [9]. This underscores the importance of fostering new forms of collaboration with industry, combining the clinical expertise of academic centres with the ability of pharmaceutical companies to register novel radiopharmaceuticals. This model represents a new way for the public and private sectors to collaborate, sharing their respective know-how and expertise.

## 3. Discussion

The implementation of EU Regulation 536/2014 has had a significant impact on academic research centres, particularly in the field of radiopharmaceuticals. While the regulation aims to harmonise clinical trial procedures across EU Member States, its stringent requirements present challenges for non-commercial institutions. Specifically, the regulation requires Good Manufacturing Practice (GMP) compliance for therapeutic radiopharmaceuticals, necessitating substantial investment in infrastructure and personnel [12].

Academic centres have had to develop new GMP-compliant facilities to continue their research activities. This process involves financial expenditure and a significant increase in the administrative workload, including preparing extensive documentation and adhering to rigorous quality control measures. Furthermore, the regulation introduces complexities in the approval process for clinical trials involving radiopharmaceuticals. Each new investigational medicinal product (IMP) requires separate authorisation, which can delay the initiation of studies and limit the flexibility of academic researchers.

The financial burden of GMP compliance is another critical issue. Substantial costs are incurred through facility construction, equipment, personnel training and ongoing quality assurance programmes. For many academic institutions, securing funding through grants and public–private partnerships are essential to sustaining these operations.

Despite these challenges, the regulation also presents opportunities for academic centres to enhance their research capabilities. Meeting GMP standards can improve the quality and reliability of clinical trials, potentially increasing competitiveness in the international research arena.

To speed up the entry of new radiopharmaceuticals into the market, the academic community could contribute by developing new targeting ligands, combinations, or radioisotopes. Basic, preclinical and translational research are particularly important for progress in this field, and there is an urgent need for academic research to receive appropriate support from pharmaceutical companies and regulatory agencies [16].

National agencies could support academic centres and academic GMP facilities by providing targeted funding for infrastructure, a network of clinical sites, and research and development projects. National agencies could also support centres by making scientific advice more accessible to non-profit organisations. Unfortunately, the regulation does not include simplified GMP pathways for low-volume or non-commercial therapeutic radiopharmaceuticals.

In this new era of radiopharmaceuticals, collaboration between clinicians and researchers from various disciplines is crucial. Chemists play a crucial role in creating more effective targeting ligands for radiopharmaceuticals. Nuclear physicists can facilitate the production of various high-purity radioisotopes with the capability for in-house generation. Alpha particles, with their short emission range and high linear energy transfer (LET), show significant promise in RPT. Finally, oncologists, experts in neurodegenerative disorders, cardiologists, and nuclear medicine doctors must collaborate to expand the clinical applications of radiopharmaceuticals across various fields [24].

In conclusion, while EU Regulation 536/2014 aims to standardise clinical trial processes, its implementation poses significant challenges for academic research centres [20]. Addressing these challenges requires a balanced approach that considers both regulatory compliance and the unmet needs of non-commercial research institutions.

## 4. Materials and Methods

### 4.1. Overview of the Regulatory Context

The first analysis was conducted from a regulatory perspective, given that regulatory aspects have been the primary drivers of change. A systematic review of the regulatory framework governing clinical trials with radiopharmaceuticals was undertaken, focusing particularly on the implementation of Regulation (EU) No. 536/2014 on clinical trials of medicinal products for human use. Additionally, European guidance documents and official publications issued by the European Commission, the European Medicines Agency (EMA) and national competent authorities were analysed. Particular attention was given to EMA guidance on investigational medicinal products (IMPs) and Good Manufacturing Practice (GMP) requirements for radiopharmaceuticals, as well as recommendations for diagnostic and therapeutic radiopharmaceuticals in a theranostic setting.

Complementing the regulatory review, a literature analysis was performed. Peer-reviewed articles, position papers from professional societies (e.g., the European Association of Nuclear Medicine (EANM)) and institutional reports addressing the regulatory environment of radiopharmaceuticals were identified via PubMed.

These sources were reviewed to capture the scientific, ethical and practical implications of the regulatory changes. A comparative assessment was then carried out between the provisions of Regulation (EU) No. 536/2014 and Directive 2001/20/EC, focusing on elements that directly influence academic and investigator-initiated clinical trials involving radiopharmaceuticals. Key areas of analysis included authorisation procedures, submission requirements through the Clinical Trials Information System (CTIS), GMP and manufacturing requirements for radiopharmaceutical IMPs and the administrative and economic burden on academic sponsors. Finally, since Regulation (EU) No. 536/2014 aims to harmonise clinical trial procedures across the European Union, the potential impact on different Member States was analysed, considering their pre-existing national frameworks before the regulation entered into force.

### 4.2. Overview of Technical Aspects

A multidisciplinary team, including pharmacists, engineers, and representatives from the Technology Transfer Office (TTO), the Research Office, and the Administrative Office, conducted the analysis. Initially, the specific items to be evaluated were defined, covering technical, economic and infrastructural aspects. The pre-GMP assessment focused on feasibility, resource requirements and potential workflow and infrastructure bottlenecks. The post-GMP evaluation considered compliance with regulatory standards, process optimisation, cost-effectiveness and scalability. Technical analyses included equipment specifications, process validation, and quality control measures. Economic analyses examined capital and operational expenditure, potential returns on investment and cost–benefit scenarios. Infrastructure assessments reviewed laboratory layout, utilities, and compatibility with GMP requirements. Each item was scored according to standardised criteria, and the results were discussed at multidisciplinary meetings to ensure consensus. Data were systematically recorded and integrated into a comprehensive report. This approach enabled critical points and optimisation strategies to be identified before GMP implementation.

### 4.3. Overview of Clinical Scenario

A literature review was performed to assess the clinical implications of the evolving regulatory framework and evaluate the impact of the implementation of Regulation (EU) No. 536/2014 on academic clinical trials involving radiopharmaceuticals, both past and present. The focus was on research into rare cancers and conditions overlooked by the pharmaceutical industry, where academic studies are essential. The research focused on peer-reviewed original research articles, reviews, editorials and policy-oriented papers published in English from 2001 onwards (i.e., after the introduction of Directive 2001/20/EC). Position papers and consensus statements from professional societies (e.g., the European Association of Nuclear Medicine (EANM), the European Society for Medical Oncology (ESMO), and national nuclear medicine societies) were also considered to capture expert perspectives on implications for clinical practice. Specific attention was given to challenges reported in academic settings, including regulatory and administrative barriers, and resource constraints. The findings from the literature were analysed to determine the potential influence of regulatory provisions on clinical research output, timelines and patient recruitment in nuclear medicine, with a particular focus on academic initiatives and research into rare tumours. Data on multidisciplinary analysis were collected from institutional reports of IRST multidisciplinary meetings, as well as from data extracted from our electronic medical record system, CCE Log 80 2.6. Additionally, we will analyse therapeutic capabilities in specialised facilities with GMP-compliant infrastructure, as these requirements could restrict the number of active sites and limit patient access, particularly in academic and non-commercial settings.

## Figures and Tables

**Figure 1 pharmaceuticals-18-01709-f001:**
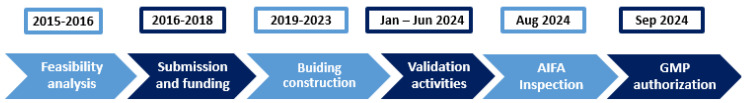
Flow chart for the activation of GMP Facility.

**Table 1 pharmaceuticals-18-01709-t001:** Activation times of IRST Facility.

Activation Items	Period
Internal analysis and dialogue with AIFA (open AIFA)	2015–2016
Submission and funding of MISE * and RER ** calls	2016–2018
Authorization process for infrastructure implementation	2019–2024 ***
Application and GMP authorization	2024
Clinical trials activation according to regulation 536	2025

* Ministry of Economic Development. ** Emilia Romagna Region. *** COVID 19 pandemic and flood in Emilia Romagna region have further slowed down the site activation of about 1 year.

**Table 2 pharmaceuticals-18-01709-t002:** Cost categories.

Cost Categories	Description	Item	Cost Proportion (% of Total Operational Budget)
Premise and Equipment	Maintenance of periodic qualification in compliance with GMP *: Annex 1 “Manufacture of Sterile Medicinal Products”, Annex 3 “Manufacture of Radiopharmaceuticals”, Annex 15 Qualification and validation, Chapter 3 “Premise and Equipment”	Increased frequency and complexity compared to hospital activities	30%
Personnel	Maintenance of organisation chart in compliance with GMP *: Chapter 2 “Personnel”, Annex 16 “Certification by a Qualified Person and Batch Release”	Increased FTE ** due to: fixed roles separated staff for Production and Quality Control	25%
Quality System	Maintenance in compliance with: Chapter 1 “Pharmaceutical Quality System” *, Chapter 4 “Documentation” * Part III * ICH guideline Q9 on quality risk management ICH guideline Q10 on pharmaceutical quality system	Increased complexity compared to health accreditation systems; Need for external GMP expertise/consultants	10%
Consumables and Supplies ***	Maintenance of Qualified Suppliers; Qualification of an additional supplier for each material (backup)	Critical in case of new tenders.	30%
Authorizations and Inspections	Payment of fees in case of variations, extensions, request or new production lines, additional QP Fixed costs for periodic inspection	Reduced flexibility compared to hospital activities	5%

* EudraLex-Volume 4-Good Manufacturing Practice (GMP) guidelines. ** Full-Time Equivalent. *** excluding the cost of the radionuclide.

**Table 3 pharmaceuticals-18-01709-t003:** Activation times for new IMPs.

IMP for New Clinical Trial	Activities Required	Estimated Time
Same IMPs * used for a new clinical trial.	No new activities	-
New IMP not yet authorised, labelled with authorised radionuclide	Need for Change according to EU GMP Part I: risk analysis, evaluation of new material and new suppliers, validation activities; new IMPD **	3–6 months
New IMP not yet authorised labelled with a new radionuclide not yet authorised	Need for an Authorization extension. Change according to EU GMP Part I: risk analysis, evaluation of new material and new suppliers, validation activities; new IMPD **	6–12 months

* IMP: Investigational Medicinal Product. ** IMPD: Investigational Medicinal Product Dossier.

## Data Availability

The original contributions presented in this study are included in the article. Further inquiries can be directed to the corresponding author.

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
