# Peer review of "How Regulation 536/2014 Is Changing Academic Research with Therapeutic Radiopharmaceuticals: A Local Experience"

_pharmaceuticals, 2025, doi:10.3390/ph18111709_

Round 1
Reviewer 1 Report
Comments and Suggestions for Authors
This manuscript provides a well-structured and informative overview of the impact of Regulation (EU) No 536/2014 on academic research involving therapeutic radiopharmaceuticals, focusing on the experience of IRST (Italy). The paper is timely, as the Regulation has just come fully into force (January 2025), and there is a lack of systematic documentation of its consequences for non-profit clinical research centers. The authors effectively combine regulatory, technical, and clinical perspectives and support their narrative with historical data and institutional experience. The manuscript is suitable for publication after minor revisions to improve clarity, scientific rigor, and contextual balance.
- The abstract could briefly include the number of clinical trials, patients, or years of experience to provide more context.
- The paper largely focuses on the IRST case study. While this is clearly stated, it would strengthen the manuscript to include a more comparative dimension—briefly summarizing how similar transitions have been managed in other EU countries (e.g., Germany, Netherlands, Spain). This could help readers assess whether IRST’s experience reflects broader European trends or unique national challenges.
- The study would benefit from more quantitative indicators where possible. For example: -Budgetary impact (approximate % of total research funding allocated to GMP implementation).-Number of radiopharmaceuticals produced annually before vs. after GMP authorization.-Comparative timelines or success rates in clinical trial activation before and after Regulation 536/2014. Even approximate ranges would make the “impact” analysis more robust.
- The “Materials and Methods” section describes a multi-disciplinary analysis but lacks specific details about how data were collected (e.g., institutional records, interviews, internal audits). Clarifying the sources of information will increase reproducibility and credibility.
- The discussion could better articulate recommendations for policymakers. For instance: How might national agencies support academic GMP compliance (grants, centralized facilities)? Should the Regulation include simplified GMP pathways for low-volume or non-commercial radiopharmaceuticals?
- The authors briefly mention ethical challenges regarding patient access and shortages. Expanding this section with examples (e.g., restricted access to Lu-177 agents, regional inequalities) and possible mitigation strategies would enrich the discussion.
- Tables 1–3 are informative but could include concise legends explaining abbreviations and details. Consider summarizing the cost table (Table 2) with approximate cost proportions (% of total operational budget).
- Reference 1 (Regulation 492/2011) seems unrelated—it should probably refer directly to Regulation (EU) No 536/2014. Please check.
- The manuscript lacks any graphical summary. A schematic flowchart of IRST’s GMP transition timeline (2015–2025) or regulatory workflow could visually enhance reader engagement.
The English is generally clear but would benefit from light editing for flow and consistency (e.g., “non-profit” vs. “no profit”; “authorisation” vs. “authorization”). Some long sentences in the introduction could be split for readability.
Author Response
Comment1: The abstract could briefly include the number of clinical trials, patients, or years of experience to provide more context.
Response1: According to reviewer suggestion, we have included these data in the abstract.
Comment2: The paper largely focuses on the IRST case study. While this is clearly stated, it would strengthen the manuscript to include a more comparative dimension—briefly summarizing how similar transitions have been managed in other EU countries (e.g., Germany, Netherlands, Spain). This could help readers assess whether IRST’s experience reflects broader European trends or unique national challenges.
Response2: This issue has been discussed in item 2.1
Comment3: The study would benefit from more quantitative indicators where possible. For example: -Budgetary impact (approximate % of total research funding allocated to GMP implementation).-Number of radiopharmaceuticals produced annually before vs. after GMP authorization.-Comparative timelines or success rates in clinical trial activation before and after Regulation 536/2014. Even approximate ranges would make the “impact” analysis more robust.
Response3: We’ve added the data on budget impact as suggested in item 2.2.4
Comment4: The “Materials and Methods” section describes a multi-disciplinary analysis but lacks specific details about how data were collected (e.g., institutional records, interviews, internal audits). Clarifying the sources of information will increase reproducibility and credibility.
Response4: We have integrated with some details about multidisciplinary data sources in item 4.3; we have also integrated in results (item 2.3) with the frequency and the number of patients discussed in each meeting in the results section.
Comment5: he discussion could better articulate recommendations for policymakers. For instance: How might national agencies support academic GMP compliance (grants, centralized facilities)? Should the Regulation include simplified GMP pathways for low-volume or non-commercial radiopharmaceuticals?
Response5: We have further elaborated in the Discussion section (item 3) on how, in our opinion, national agencies could support academic centers through regulatory and financial assistance. Unfortunately the Regulation doesn’t include any simplification of GMP pathways for low-volume or non-commercial therapeutic investigational radiopharmaceuticals.
Comment6: The authors briefly mention ethical challenges regarding patient access and shortages. Expanding this section with examples (e.g., restricted access to Lu-177 agents, regional inequalities) and possible mitigation strategies would enrich the discussion
Response6: This is a very critical point that we have tried to expand at item 2.3.
Comment7: Tables 1–3 are informative but could include concise legends explaining abbreviations and details. Consider summarizing the cost table (Table 2) with approximate cost proportions (% of total operational budget).
Response7: We’ve integrated the Table 2 with cost proportions for each categories and added legends in the three tables
Comment8: Reference 1 (Regulation 492/2011) seems unrelated—it should probably refer directly to Regulation (EU) No 536/2014. Please check.
Response8: We’ve corrected the reference
Comment9: The manuscript lacks any graphical summary. A schematic flowchart of IRST’s GMP transition timeline (2015–2025) or regulatory workflow could visually enhance reader engagement
Response9: We’ve added a schematic flow chart as suggested
Comment: The English is generally clear but would benefit from light editing for flow and consistency (e.g., “non-profit” vs. “no profit”; “authorisation” vs. “authorization”). Some long sentences in the introduction could be split for readability
Response: We’ve tried to improve English
Reviewer 2 Report
Comments and Suggestions for Authors
I appreciated your paper. I believe that the structure of your article is robust and that all the factors necessary for the implementation of the new regulation have been analyzed and commented on in detail.
I believe that the analysis of these factors can be useful for all those who have to face this new regulation.
Author Response
Comment: I appreciated your paper. I believe that the structure of your article is robust and that all the factors necessary for the implementation of the new regulation have been analyzed and commented on in detail.
I believe that the analysis of these factors can be useful for all those who have to face this new regulation.
Response: We would like to thank the reviewer for the comments.
Reviewer 3 Report
Comments and Suggestions for Authors
This paper discusses the structural transformation brought about by the EU Regulation 536/2014 in the field of therapeutic radiopharmaceuticals. In particular, it focuses on an issue of great concern to the academic community: the mandatory GMP compliance requirement, which has led academic institutions to gradually withdraw from in-house production of therapeutic radiopharmaceuticals. This shift could seriously affect research progress in the field and have a direct impact on clinical treatment for patients with rare diseases who require highly personalized therapies. On the positive side, the regulation represents an important step toward standardization and safety in radiopharmaceutical manufacturing and clinical use. However, reasonable solutions are still needed to balance strict regulation with the need to sustain academic innovation and patient access. The authors provide valuable insights and practical perspectives that are meaningful for both researchers and regulatory authorities in this field. I believe the paper fits well with the journal’s scope and the selected Special Issue and recommend it for publication.
Author Response
Comment: The authors provide valuable insights and practical perspectives that are meaningful for both researchers and regulatory authorities in this field. I believe the paper fits well with the journal’s scope and the selected Special Issue and recommend it for publication.
Response: We would like to thank the reviewer for the comments.